# Heterotypic Supramolecular Hydrogels Formed by Noncovalent Interactions in Inflammasomes

**DOI:** 10.3390/molecules26010077

**Published:** 2020-12-26

**Authors:** Adrianna N. Shy, Huaimin Wang, Zhaoqianqi Feng, Bing Xu

**Affiliations:** Department of Chemistry, Brandeis University, 415 South St., Waltham, MA 02453, USA; ashy30@brandeis.edu (A.N.S.); wanghuaimin@westlake.edu.cn (H.W.); zfeng@brandeis.edu (Z.F.)

**Keywords:** peptides, self-assembly, hydrogels

## Abstract

The advance of structural biology has revealed numerous noncovalent interactions between peptide sequences in protein structures, but such information is less explored for developing peptide materials. Here we report the formation of heterotypic peptide hydrogels by the two binding motifs revealed by the structures of an inflammasome. Specifically, conjugating a self-assembling motif to the positively or negatively charged peptide sequence from the ASCPYD filaments of inflammasome produces the solutions of the peptides. The addition of the peptides of the oppositely charged and complementary peptides to the corresponding peptide solution produces the heterotypic hydrogels. Rheology measurement shows that ratios of the complementary peptides affect the viscoelasticity of the resulted hydrogel. Circular dichroism indicates that the addition of the complementary peptides results in electrostatic interactions that modulate self-assembly. Transmission electron microscopy reveals that the ratio of the complementary peptides controls the morphology of the heterotypic peptide assemblies. This work illustrates a rational, biomimetic approach that uses the structural information from the protein data base (PDB) for developing heterotypic peptide materials via self-assembly.

## 1. Introduction

Peptides, as the building blocks for supramolecular assemblies, have received considerable research attention [1,2,3,4,5,6,7] because peptides are biodegradable and easily accessible. Generally, when the concentration of a self-assembling peptide reaches a certain threshold, intermolecular noncovalent interactions would result in a network of peptide nanofibers, which are able to hold water in the interstitial spaces to form hydrogels [8,9]. Because of their similarity with soft tissues, peptide-based hydrogels promise many useful applications, such as tissue engineering [10,11,12,13,14,15,16,17], drug delivery [18,19,20,21,22], and cancer therapy [23,24,25,26,27]. The formation of peptide hydrogels usually originates from interactions on three levels: primary (the atomic interactions between molecules), secondary (the interactions between two or more monomeric structures to create ribbons, rods, etc.), and tertiary (the overall structure of the gel determined by individual aggregates) [8]. The assembly of these peptides by these three categories of interactions determine the overall characteristics of peptide-based hydrogels. These three types of interactions can originate from one type of building block (i.e., homotypic) or different types of building blocks (i.e., heterotypic). While most of the studies have focused on the hydrogels made of one type of peptide building blocks, that is, homotypic hydrogels, the exploration on heterotypic peptide hydrogels is rather limited [28,29,30,31].

Heterotypic peptide hydrogels, resulting from the self-assembly of peptide building blocks that bear different sequences, should increase the functional diversity of hydrogels, as shown in the case of the hydrogel for detoxifying uranium and promoting wound healing [32]. Although it is well-established that noncovalent interactions such as hydrogen bonding, hydrophobic interactions, and electrostatic interactions, play major roles [33] in the process of peptide self-assembly, it is rather difficult to determine which different sequences of short peptides would result in strong intermolecular interactions to drive self-assembly for forming heterotypic peptide hydrogels. Recently, we used small peptides derived from protein domains to form heterotypic hydrogels [30,34,35]. In those studies, the main intermolecular interactions are hydrophobic interactions and hydrogen bonding. Another type of important noncovalent interaction for self-assembly, obviously, is electrostatic interactions, which have been explored for the co-assembly of peptide amphiphiles [36]. Inspired by that result and the development of heterotypic hydrogels, we decided to use the structural information in the protein data bank (PDB) for generating heterotypic hydrogels that consist of oppositely charged and complementary peptides.

In this case we turn to the protein structure of the inflammasome, a multiprotein complex in the intracellular space, which detects pathogenic microorganisms and sterile stressors and activates inflammatory cytokines [37,38]. We used the sequences ^21^KKFKLKL^27^ and ^48^DALDLTD^54^, from the ASC^PYD^ filaments derived from the inflammasome [39,40] to enable heterotypic self-assembly. These two sequences bear opposite charges and are complementary with each other in the crystal structures of the inflammasome. We attached the well-known self-assembling motif Nap-FF [33] to the N-terminal of the peptides to generate Nap-FFKKFKLKL and Nap-FFDALDLTD. We also attached an acetyl (Ac) group in place of Nap-FF to generate Ac-KKFKLKL and Ac-DALDLTD for modulating hydrophobic interaction during the heterotypic self-assembly. Our results show that: (i) while the mixing of Nap-FFDALDLTD with Ac-KKFKLKL affords precipitates initially, some of them turn to hydrogels over time; (ii) the mixing of Nap-FFKKFKLKL with Ac-DALDLTD or Nap-FFDALDLTD results in hydrogels; (iii) the mixing of Ac-DALDLTD with Ac-KKFKLKL produces a solution and is unable to form a hydrogel. The ratio of the complementary peptides also affects the rheological properties of the resulting hydrogels. Moreover, circular dichroism indicates that the addition of the complementary peptides promotes self-assembly. In addition, transmission electron microscopy (TEM) reveals that the ratio of the oppositely charged complementary peptides controls the morphology of the heterotypic peptide assemblies. This study, illustrating the combination of self-assembling, hydrophobic peptide motif, and the complementary charged peptides (from PDB) for generating heterotypic peptide hydrogels, provides a general, biomimetic approach to design peptide soft materials with multiple components for new functions. 

## 2. Results and Discussion

Scheme 1 below shows the molecular structures of the four peptides used in this study. Drawing inspiration from the protein structures in the work of Wu et al., in which they explored the assembly of ASC dependent (apoptosis-associated speck-like protein containing a CARD) inflammasomes AIM2 and NLRP3 [40], we used two peptide sequences from the ASC^PYD^ filaments derived from the inflammasome (PDB ID: 3J63) to enable heterotypic self-assembly. The peptide KKFKLKL comes from a positively charged alpha helix, and the peptide DALDLTD is its negatively charged alpha helical counterpart (Appendix A). **1** and **3** are the positive and negatively charged peptides KKFKLKL and DALDLTD, respectively, with an acetyl group attached at the N-terminal. To promote the self-assembly of the peptides, we also attached Nap-FF, a well-established self-assembling motif [33], to the N-terminal of KKFKLKL and DALDLTD to generate **2** and **4**. Using solid phase peptide synthesis (Scheme S1), we obtained the desired peptides (**1** to **4**) in good yields.

After synthesizing the peptides, we then tested the formation of hydrogels by mixing the oppositely charged peptides. Figure 1 shows the solution of Nap-FFDALDLTD (**4**) at 0.4 wt% concentration mixed with the solutions of Ac-KKFKLKL (**1**) at different 0.5, 1, 2, 3, and 4 molar equivalents. After the mixing, all mixtures form solutions after 24 h with different turbidity. For example, while mixing of 0.5 molar equiv. of **1** with 0.4 wt% of **4** results in a slightly turbid mixture, mixing of 3 or 4 molar equiv. of **1** with 0.4 wt% of **4** results in a white suspension. That is, the amount of precipitate in the mixtures increases with increasing molar ratio of **1** versus **4**. These results are in line with the expected the electrostatic interactions between **1** and **4**. As shown in Figure 2, after five days of mixing, the same mixtures shown in Figure 1 turn into hydrogels or viscous solutions. Specifically, the mixtures of **1** and **4**, at the ratios of 1:2 and 1:1, produce a viscous liquid and a hydrogel, respectively. Further increasing the amount of **1** in the mixture apparently slows down the conversion of the precipitates to hydrogels, resulting in more phase separation to form the mixtures containing opaque precipitates and transparent hydrogels. These results indicate the 1:1 ratio of **1** and **4** is most favorable to form the heterotypic hydrogel via thermodynamic equilibrium, agreeing with the reversibility of electrostatic interactions between **1** and **4** and the 1:1 ratio of KKFKLKL and DALDLTD in the structure of the inflammasome [40]. It takes five days to form the hydrogels indicating that the molecules in the hydrogels are more ordered than in the precipitates, and the reversible electrostatic interactions allows the precipitate to transform to hydrogels, which are thermodynamically more stable.

Figure 3 shows the mixtures of Nap-FFKKFLKL (**2**) (0.4 wt%) with AcDALDLTD (**3**) at the same molar ratios found in Figure 1 and Figure 2. Unlike the results shown in Figure 1, these mixtures form gels at all ratios in the first 24 h. While **2** alone remains a solution, all the mixtures at concentrations of 0.5, 1, 2, 3, and 4 molar equiv. of **3** form the gels that exhibit slightly different rigidity. Distinctively, at 2, 3, and 4 equiv. of **3**, the gels flow slightly as indicated by the slight tilt in the gel (Figure 3D–F). The ratios 1:0.5 and 1:1 of **2** and **3** results in the most stable gels with little flow once the vial is laid on its side. The physical differences between the hydrogels in Figure 3 show that **2** as the self-assembling peptide and **3** as its negatively charged binding partner create more cohesive pairs than that of **1** and **4**. The formation of heterotypic hydrogels regardless of the ratios of **2** and **3** indicates that the conjugation of Nap-FF to KKFKLKL provide higher self-assembling propensity than the conjugation of Nap-FF to DALDLTD, suggesting that the extra phenylalanine in **2** substantially promotes self-assembly of **2** and the pair of **2** and **3**.

We also mixed **1** and **3** or **2** and **4** at the 1:1 molar ratio. As summarized in Table 1, while mixing **1** and **3** produces the solutions, mixing **2** and **4** at a 1:1 molar ratio results in a precipitate/viscous sol mixture (Appendix A) in which the precipitate forms immediately upon mixing. When 3 or 4 equiv. of **4** mixes with **2** (0.4 wt%), a gel containing precipitates forms after 27 days (Appendix A). These results indicate that it is necessary to incorporate adequate self-assembling, hydrophobic forces (provided by Nap-FF in this case) to generate the heterotypic hydrogels. But excessive hydrophobic interactions plus charge interactions favor precipitation at the initial mixing. In addition, mixing the peptide pairs with the same charge (**1** and **2** or **3** and **4**) only results in a solution, agreeing with that multiple same charges cause repulsion between the two peptides and disfavor the self-assembly.

Because mixing **2** and **3** gives hydrogels at all ratios tested, we chose the hydrogels formed by **2** and **3** for rheology measurement. Figure 4A shows the frequency dependence of G’ (storage moduli) and G” (loss modulus) of **2** with 0.5, 1, 2, 3 and 4 equiv. of **3**. For these mixtures, all G’ values dominate the corresponding G” values, agreeing that more solid-like structures are components of the gels. This trend correlates with Figure 3 above where all mixtures are seen as gels with stability, even after 24 h. To gain more information about the durability and strength of these gels, we also tested the storage and loss modulus of the mixtures of **2** and **3** under increasing amounts of stress (Figure 4B). Until a critical strain of about 9% is reached, G’ is over 5 times higher than G”, confirming that the mixtures act as hydrogels. With the increase of the amount of **3**, while the storage moduli decrease slightly, the critical strains appear to increase slightly. This result suggests that the addition of **3** is able to prevent the disruption of the gel network, though the improvement is relatively small.

To further understand the hydrogels formed by mixing **2** and **3**, we also examined the circular dichroism (CD) of the hydrogels made of **2** and **3** (Figure 5 and Appendix A). The CD signal of **3** changes with the increase of concentrations (Figure 5A). At the concentration of 1.5 mM (or 0.5 molar equivalent of **2** at 0.4 wt%), **3** exhibits two troughs at around 205 and 230 nm in the CD spectrum, indicating helical conformation and agreeing with the conformation of DALDLTD in the crystal structure. Increasing the concentrations of **3** results in a more negative trough around 230 nm, while the trough around 205 nm becomes shallow. This change suggests the intermolecular interactions among the molecules of **3** while **3** remain largely in a helical conformation. NapFFKKFKLKL (Figure 5B) at the concentration of 0.4 wt% shows two negative troughs at 203 nm and 222 nm, agreeing with the helical secondary structures of KKFKLKL observed in the crystal structure. The intense positive peak at 243 nm likely is the induced CD signal that originated from the naphthyl group. The mixtures of peptides **2** and **3** at all equivalent concentrations (Figure 5C) show shallow troughs at around 203 and 222 nm. This observation agrees with the antiparallel dipolar moments of C=O bonds resulted from the charge pairing of **2** and **3** in the self-assembly. The most prominent changes in the CD spectra is the sign reversal of the induced CD signal at around 243 nm. This result confirms the interaction of **2** and **3**. When 4 equivalents of **3** mix with **2**, the induced CD signal of naphthyl group also blue shifts slightly, indicating that the excess **3** dilutes the pair of **2** and **3** and decreases the self-assembly, which also agrees with the decrease of the storage modulus of the hydrogels with the increase of the concentrations of **3**. Fluorescent spectra (Appendix A) of the mixtures of **2** and **3** show that the fluorescence from naphthyl groups increases after the additions of 0.5, 1, 2, and 3 equivalents of AcDALDLTD (**3**) into Nap-FFKKFKLKL (**2**), agreeing with that the interactions between **2** and **3** would separate the naphthyl group to minimize self-quenching. Conversely, adding 4 equivalents of **3** into **2** results a slight decrease of fluorescent intensity, suggesting excess negative charges likely quench the fluorescence from the naphthyl group. There results support the intermolecular interactions between **2** and **3**.

Figure 6 displays a range of morphologies from loosely packed nanoparticles to nanoribbons as the equivalents of **1** increase versus **4**. Figure 6A is the TEM image of **4** at 0.4 wt%, which shows nanoparticles. The naphthalene rings on **4** likely promotes self-assembly, but repulsion by the negatively charged peptide residues disfavor extended nanostructures, such as nanofibers. Figure 6B shows nanoparticles that are less disperse in the mixture of **1** and **4** in a 0.5:1 ratio, indicating that **1** interacts with **4**. At a 1:1 molar ratio of **4** and **1**, the mixture displays a mixture of nanoribbons and particles (Figure 6C). It is also the only combination that forms a homogenous hydrogel (Figure 2C). Interestingly, as the ratio of **1** to **4** increases the nanoribbons become more apparent (Figure 6D,E), although their corresponding optical images are turbid viscous gel-like solutions. The turbidity may be due to these nanoribbons coalescing at the bottom of the vial forming a gel while the excess water separates from the gel network (i.e., syneresis due to interfibrillar interactions). Figure 6F shows the nanoribbons only, agreeing with Figure 2F which shows a gel/sol mixture. These observations suggest that excess amount of **4** likely remains in the solution to drive the equilibrium to form the 1:1 complex of **1** and **4**, which self-assembles to form the nanoribbons.

Figure 7 shows the TEM of the solution and hydrogels shown in Figure 3. Figure 7A shows very thin nanofibers, with a diameter of 15 ± 2 nm, indicating that **2** already self-assembles at the concentration of 0.4 wt% in water. These thin nanofibers likely repel each other, suggesting that **2** alone is a fluid. At the 1:0.5 mixture of **2** and **3**, TEM reveals densely packed nanofibers with considerable entanglement (Figure 7B), agreeing with the formation of the hydrogel (Figure 3B). When **2** and **3** are in a 1:1 ratio, TEM displays nanofibers that are woven together in a twisted manner (Figure 7C), suggesting the alternating arrangements of **2** and **3**, as shown in the crystal structures of inflammasomes. With the increase of the amount of **3**, the nanofibers largely maintain the same pattern; however, there are more braided-like bundles of nanofibers accompanied by increasing pore size of the network. This observation agrees with the slight decrease of the viscoelasticity of the hydrogels when the equivalence of **3** increases from 2 to 4. 

Figure 8A shows the TEM image of 0.25 wt% of **2**, which is observed to have an interwoven network of nanofibers with the diameter of 14 ± 2 nm, which is similar to the nanofibers formed in the solution of 0.4 wt% of **2** (Figure 7A). The longer nanofibers at 0.25 wt% than at 0.4 wt% of **2** indicates weaker intrafiber repulsion between the molecules of **2** at 0.25 wt% than at 0.4 wt%. For the solution of **4** at 0.25 wt%, TEM also shows nanoparticles (Figure 8B), being less dense than those in 0.4 wt% of **4** (Figure 7B). When 0.25 wt% of **2** and 0.25 wt% of **4** are mixed, TEM shows long and thick nanofibers coated with nanoparticles, but with quite low density. These results agree with the observation that **2** or **4** or their initial mixtures is a solution. When the concentration of the negatively charged peptide increases to 3 and 4 molar equivalents of **4** in the mixture of **2** and **4** and after the viscous solutions, sitting for 27 days, become gels, TEM reveals dense networks of fibers (Appendix A). Additionally, 4 equivalents of **4** in the mixture of **2** and **4** results in thinner nanofibers, suggesting that the excess amount of **4** provides electrostatic repulsions. The observation suggests that the ratio of the components in the heterotypic hydrogels is a useful handle for tuning molecular self-assembly.

## 3. Materials and Methods

### 3.1. Materials

#### 3.1.1. Procedure for Hydrogel Preparation

Nap-FFKKFKLKL (5.0 mg) was dissolved in 1 mL of aqueous buffer (sodium hydroxide was used to adjust the final pH to 7.4), the hydrogels were formed after the addition of different equivalents of other peptides (the initial concentration was 100 mM), and the final concentration of Nap-FFKKFKLKL is 0.4 wt% in all hydrogels. The gelation was determined by using the vial inversion method.

#### 3.1.2. Circular Dichroism Measurement

CD spectra were recorded (185−300 nm) using a JASCO 810 spectrometer under a nitrogen atmosphere. The hydrogel (0.4 wt%, 200 μL) was placed evenly on the 1 mm thick quartz cuvette and scanned with a 0.1 nm interval three times.

#### 3.1.3. Rheology Experiment

The rheological tests were performed on a TA ARES-G2 rheometer with parallel-plate geometry and an upper plate diameter of 25 mm, while the gap was 0.4 mm. During the measurement, the stage temperature was maintained at 25 °C by a Peltier heating cooling system. The hydrogel was loaded into stage by spatula, and then we performed time sweep at the frequency of 6.28 rad/s and the strain for 1.0%. The frequency sweep was performed at the range of 0.1 to 200 rad/s and the strain was 1.0%.

#### 3.1.4. TEM Sample Preparation 

1. Place hydrogel on the grid (5 µL, sufficient to cover the grid surface).

2. Rinsing: ~30 s later, place a large drop of the ddH_2_O on parafilm and let the grid touch the water drop, with the sample-loaded surface facing the parafilm. Tilt the grid and gently absorb water from the edge of the grid using a filter paper sliver. (3 times)

3. (immediately after rinsing) Staining: place a large drop of the UA (uranyl acetate, 2%) stain solution on parafilm and let the grid touch the stain solution drop, with the sample-loaded surface facing the parafilm. Tilt the grid and gently absorb the stain solution from the edge of the grid using a filter paper sliver. 

4. Allow the grid to dry in air and examine the grid as soon as possible.

### 3.2. Instruments

Rheological tests were conducted on a TA ARES-G2 rheometer at 25 ℃. Circular Dichroism (CD) spectra were obtained by a JASCO J-810 spectropolarimeter (JASCO Inc., Easton, PA, USA). Transmission Electron Microscopy (TEM) staining was done using 2% uranyl acetate and images were taken on a Morgagni 268 transmission electron microscope (FEI, Hillsboro, OR, USA).

## 4. Conclusions

By using oppositely charged, complimentary peptides from the inflammasome, we have shown various characteristics of heterotypic self-assembly. This work further establishes that protein structures are a valuable source for developing biomimetic, heterotypic peptide hydrogels. It also reveals that kinetics plays an important role in forming heterotypic hydrogels made of components that bear opposite charges and hydrophobic interactions. Carefully combining electrostatic and hydrophobic interactions, with additional consideration of kinetics, likely would be a facile approach to tailor the properties of the heterotypic hydrogels for applications. This investigation may offer a starting point for the use of electrostatically assembled hydrogels to achieve various purposes. In the future, exploring the characteristics of three or more different peptides from other protein structures may be useful in understanding protein aggregation from different diseases, thus contributing to the development of therapeutics.

## Data Availability

The data presented in this study are available on request from the corresponding author.

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
