# Peer review of "Heterotypic Supramolecular Hydrogels Formed by Noncovalent Interactions in Inflammasomes"

_molecules, 2020, doi:10.3390/molecules26010077_

Round 1

Reviewer 1 Report

The manuscript reports on the fabrication of heterotypic peptide hydrogels using Nap-FFKKFKLKL and Ac-DALDLTD. Four peptides containing KKFKLKL or DALDLTD, and Nap-FF or acetyl terminals, respectively, were synthesized and tested. The work showed some interesting results on the formation of peptide fibrils and hence the hydrogels. However, there are some points that need to be clarified before the manuscript can be considered for publication.

  1. The interactions among the peptides were not very well characterized. Although electrostatic force may be a main interaction that drives the self-assembly, the function of the terminal groups, i.e. Nap-FF and acetyl, was not well-demonstrated. Only CD and TEM investigations are not enough.
  2. The kinetics should be further studied by at least rheology characterization.
  3. The content in the Materials and Methods section is too brief. The preparation of the hydrogels/samples and the procedures for the instrumental characterizations should be carefully described with necessary details.
  4. The manuscript needs to be carefully checked. For example in Figure 4, it mentions “without or with the addition of 3”. However, it seems that the curves only involve the rheology data with AcDALDLTD (3).

Reviewer 2 Report

This manuscript described heterotypic supramolecular hydrogel formations through noncovalent interactions found in inflammasomes. Although charge complementarity of positively and negatively charged peptides to form self-assembled structures and/or hydrogels might be very novel, but the described biomimetic strategy could be beneficial for the readers. The manuscript was written clearly. I would like to recommend publication of this manuscript in molecules. Minor points that I would like to ask are as follows.

-It would be useful to know amino acid numbers of the peptides derived from inflammasome and the 3D structure of inflammasome.

-Figure 5: It would be better to include HT values for CD spectra. Also, it would be more comprehensible when the spectra were shown in the same wavelength range.

Round 2

Reviewer 1 Report

The manuscript has been properly revised by the authors and therefore is recommended for publication.

Author Response

We thanks the reviewer for the suggestion. We have go over the manuscript to correct the minor grammatical errors.